# Machine Comprehension Using Match-LSTM and Answer Pointer

**Shuohang Wang**
School of Information Systems
Singapore Management University
shwang.2014@phdis.smu.edu.sg

**Jing Jiang**
School of Information Systems
Singapore Management University
jingjiang@smu.edu.sg

## Abstract

Machine comprehension of text is an important problem in natural language processing. A recently released dataset, the Stanford Question Answering Dataset (SQuAD), offers a large number of real questions and their answers created by humans through crowdsourcing. SQuAD provides a challenging testbed for evaluating machine comprehension algorithms, partly because compared with previous datasets, in SQuAD the answers do not come from a small set of candidate answers and they have variable lengths. We propose an end-to-end neural architecture for the task. The architecture is based on match-LSTM, a model we proposed previously for textual entailment, and Pointer Net, a sequence-to-sequence model proposed by Vinyals et al. (2015) to constrain the output tokens to be from the input sequences. We propose two ways of using Pointer Net for our tasks. Our experiments show that both of our two models substantially outperform the best results obtained by Rajpurkar et al. (2016) using logistic regression and manually crafted features. Besides, our boundary model also achieves the best performance on the MSMARCO dataset (Nguyen et al., 2016).

## 1 Introduction

Machine comprehension of text is one of the ultimate goals of natural language processing. While the ability of a machine to understand text can be assessed in many different ways, in recent years, several benchmark datasets have been created to focus on answering questions as a way to evaluate machine comprehension (Richardson et al., 2013; Hermann et al., 2015; Hill et al., 2016; Weston et al., 2016; Rajpurkar et al., 2016; Nguyen et al., 2016). In this setup, typically the machine is first presented with a piece of text such as a news article or a story. The machine is then expected to answer one or multiple questions related to the text.

In most of the benchmark datasets, a question can be treated as a multiple choice question, whose correct answer is to be chosen from a set of provided candidate answers (Richardson et al., 2013; Hill et al., 2016). Presumably, questions with more given candidate answers are more challenging. The Stanford Question Answering Dataset (SQuAD) introduced recently by Rajpurkar et al. (2016) contains such more challenging questions whose correct answers can be any sequence of tokens from the given text. Moreover, unlike some other datasets whose questions and answers were created automatically in Cloze style (Hermann et al., 2015; Hill et al., 2016), the questions and answers in SQuAD were created by humans through crowdsourcing, which makes the dataset more realistic. Another real dataset, the Human-Generated MAchine Reading COmprehension dataset (MSMARCO) (Nguyen et al., 2016), provided a query together with several related documents collected from Bing Index. The answer to the query is generated by human and the answer words can not only come from the given text.

Given these advantages of the SQuAD and MSMARCO datasets, in this paper, we focus on these new datasets to study machine comprehension of text. A sample piece of text and three of its associated questions from SQuAD are shown in Table 1. Traditional solutions to this kind of question answering tasks rely on NLP pipelines that involve multiple steps of linguistic analyses and feature engineering, including syntactic parsing, named entity recognition, question classification, semantic parsing, etc. Recently, with the advances of applying neural network models in NLP, there has been

---

In 1870, Tesla moved to Karlovac, **to attend school at the Higher Real Gymnasium**, where he was profoundly influenced by a math teacher **Martin Sekulić**. The classes were held in **German**, as it was a school within the Austro-Hungarian Military Frontier. Tesla was able to perform integral calculus in his head, which prompted his teachers to believe that he was cheating. He finished a four-year term in three years, graduating in 1873.

| | |
|---|---|
| 1. In what language were the classes given? | German |
| 2. Who was Tesla's main influence in Karlovac? | Martin Sekulić |
| 3. Why did Tesla go to Karlovac? | attend school at the Higher Real Gymnasium |

Table 1: A paragraph from Wikipedia and three associated questions together with their answers, taken from the SQuAD dataset. The tokens in bold in the paragraph are our predicted answers while the texts next to the questions are the ground truth answers.

much interest in building end-to-end neural architectures for various NLP tasks, including several pieces of work on machine comprehension (Hermann et al., 2015; Hill et al., 2016; Yin et al., 2016; Kadlec et al., 2016; Cui et al., 2016). However, given the properties of previous machine comprehension datasets, existing end-to-end neural architectures for the task either rely on the candidate answers (Hill et al., 2016; Yin et al., 2016) or assume that the answer is a single token (Hermann et al., 2015; Kadlec et al., 2016; Cui et al., 2016), which make these methods unsuitable for the SQuAD/MSMARCO dataset. In this paper, we propose a new end-to-end neural architecture to address the machine comprehension problem as defined in the SQuAD/MSMARCO dataset. And for the MSMARCO dataset, we will only make use of the words in the given text to generate the answer.

Specifically, observing that in the SQuAD/MSMARCO dataset many questions could be entailed from some sentences in the original text, we adopt a match-LSTM model that we developed earlier for textual entailment (Wang & Jiang, 2016) as one layer of our model. We build a bi-directional match-LSTM on the given passage with attentions on the question for each word so that each position in the paragraph will have a hidden representation reflecting its relation to the question. Then we further adopt the Pointer Net (Ptr-Net) model developed by Vinyals et al. (2015) to select the words in these positions based on the hidden representations built by match-LSTM as an answer. We propose two ways to apply the Ptr-Net model for our task: a **sequence model** which selects the answer word by word, and a **boundary model** which only selects the start and end points of the answer span. Experiments on the SQuAD dataset show that our two models both outperform the best performance reported by Rajpurkar et al. (2016). Moreover, using an ensemble of several of our models, we can achieve very competitive performance on SQuAD. For the MSMARCO dataset, a real query based problem, our boundary model outperforms our sequence model with a big margin. It also outperforms the golden passage baseline.

Our contributions can be summarized as follows: (1) We propose two new end-to-end neural network models for machine comprehension, which combine match-LSTM and Ptr-Net to handle the special properties of the SQuAD dataset. To the best of our knowledge, we are the first to propose the boundary model which is more suitable to the SQuAD/MSMARCO tasks. And we are the first to integrate the attention-based word pair matching into machine comprehension tasks. (2) We have achieved the performance of an exact match score of 71.3% and an F1 score of 80.8% on the unseen SQuAD test dataset, which is much better than the feature-engineered solution (Rajpurkar et al., 2016). Our performance is also close to the state of the art on SQuAD, which is 74.8% in terms of exact match and 82.2% in terms of F1 collected from the SQuAD Leaderboard [1]. Besides, our boundary model achieves the state-of-art performance on the MSMARCO dataset with BLUE-1/2/3/4 40.7/33.9/30.6/28.7 and Rouge-L 37.3 [2]. (3) Our further visualization of the models reveals some useful insights of the attention mechanism for reasoning the questions. And we also show that the boundary model can overcome the early stop prediction problem in the sequence model. Besides, we also made our code available online [3].

---

[1] https://rajpurkar.github.io/SQuAD-explorer/

[2] http://www.msmarco.org/leaders.aspx

[3] https://github.com/shuohangwang/SeqMatchSeq

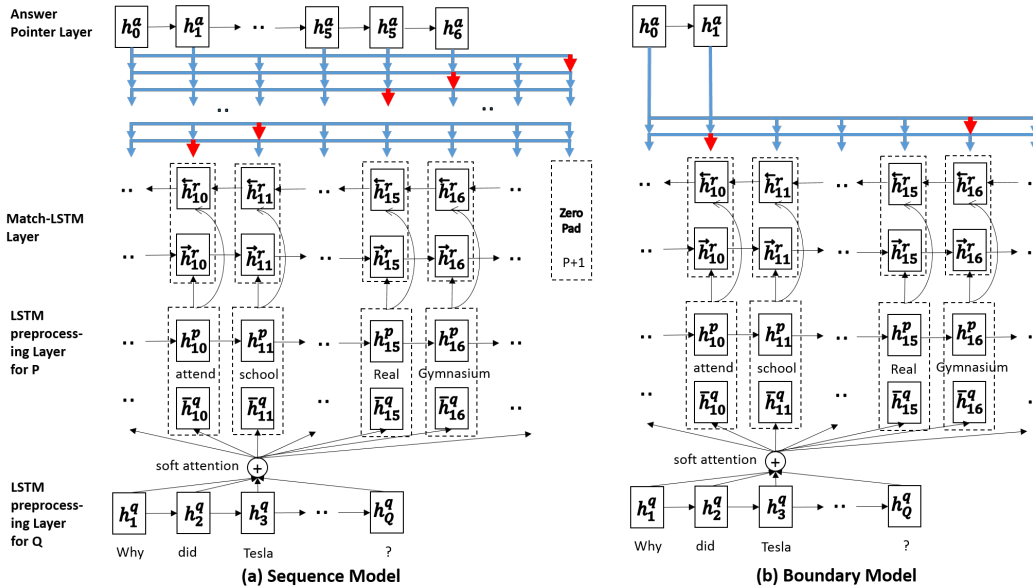

Figure 1: An overview of our two models. Both models consist of an LSTM preprocessing layer, a match-LSTM layer and an Answer Pointer layer. For each match-LSTM in a particular direction, $\bar{h}_i^q$, which is defined as $\mathbf{H}^q \alpha_i^\mathsf{T}$, is computed using the $\alpha$ in the corresponding direction, as described in Eqn. (2)

## 2 METHOD

In this section, we first briefly review match-LSTM and Pointer Net. These two pieces of existing work lay the foundation of our method. We then present our end-to-end neural architecture for machine comprehension.

### 2.1 MATCH-LSTM

In a recent work on learning natural language inference, we proposed a match-LSTM model for predicting textual entailment (Wang & Jiang, 2016). In textual entailment, two sentences are given where one is a premise and the other is a hypothesis. To predict whether the premise entails the hypothesis, the match-LSTM model goes through the tokens of the hypothesis sequentially. At each position of the hypothesis, attention mechanism is used to obtain a weighted vector representation of the premise. This weighted premise is then to be combined with a vector representation of the current token of the hypothesis and fed into an LSTM, which we call the match-LSTM. The match-LSTM essentially sequentially aggregates the matching of the attention-weighted premise to each token of the hypothesis and uses the aggregated matching result to make a final prediction.

### 2.2 POINTER NET

Vinyals et al. (2015) proposed a Pointer Network (Ptr-Net) model to solve a special kind of problems where we want to generate an output sequence whose tokens must come from the input sequence. Instead of picking an output token from a fixed vocabulary, Ptr-Net uses attention mechanism as a pointer to select a position from the input sequence as an output symbol. The pointer mechanism has inspired some recent work on language processing (Gu et al., 2016; Kadlec et al., 2016). Here we adopt Ptr-Net in order to construct answers using tokens from the input text.

### 2.3 OUR METHOD

Formally, the problem we are trying to solve can be formulated as follows. We are given a piece of text, which we refer to as a passage, and a question related to the passage. The passage is

represented by matrix $\mathbf{P} \in \mathbb{R}^{d \times P}$, where $P$ is the length (number of tokens) of the passage and $d$ is the dimensionality of word embeddings. Similarly, the question is represented by matrix $\mathbf{Q} \in \mathbb{R}^{d \times Q}$ where $Q$ is the length of the question. Our goal is to identify a subsequence from the passage as the answer to the question.

As pointed out earlier, since the output tokens are from the input, we would like to adopt the Pointer Net for this problem. A straightforward way of applying Ptr-Net here is to treat an answer as a sequence of tokens from the input passage but ignore the fact that these tokens are consecutive in the original passage, because Ptr-Net does not make the consecutivity assumption. Specifically, we represent the answer as a sequence of integers $\mathbf{a} = (a_1, a_2, \ldots)$, where each $a_i$ is an integer between 1 and $P$, indicating a certain position in the passage.

Alternatively, if we want to ensure consecutivity, that is, if we want to ensure that we indeed select a subsequence from the passage as an answer, we can use the Ptr-Net to predict only the start and the end of an answer. In this case, the Ptr-Net only needs to select two tokens from the input passage, and all the tokens between these two tokens in the passage are treated as the answer. Specifically, we can represent the answer to be predicted as two integers $\mathbf{a} = (a_s, a_e)$, where $a_s$ an $a_e$ are integers between 1 and $P$.

We refer to the first setting above as a *sequence* model and the second setting above as a *boundary* model. For either model, we assume that a set of training examples in the form of triplets $\{(\mathbf{P}_n, \mathbf{Q}_n, \mathbf{a}_n)\}_{n=1}^{N}$ are given.

An overview of the two neural network models are shown in Figure 1. Both models consist of three layers: (1) An LSTM preprocessing layer that preprocesses the passage and the question using LSTMs. (2) A match-LSTM layer that tries to match the passage against the question. (3) An Answer Pointer (Ans-Ptr) layer that uses Ptr-Net to select a set of tokens from the passage as the answer. The difference between the two models only lies in the third layer.

**LSTM Preprocessing Layer**

The purpose for the LSTM preprocessing layer is to incorporate contextual information into the representation of each token in the passage and the question. We use a standard one-directional LSTM (Hochreiter & Schmidhuber, 1997) to process the passage [4] and the question separately, as shown below:

$$\mathbf{H}^p = \overrightarrow{LSTM}(\mathbf{P}), \quad \mathbf{H}^q = \overrightarrow{LSTM}(\mathbf{Q}). \tag{1}$$

The resulting matrices $\mathbf{H}^p \in \mathbb{R}^{l \times P}$ and $\mathbf{H}^q \in \mathbb{R}^{l \times Q}$ are hidden representations of the passage and the question, where $l$ is the dimensionality of the hidden vectors. In other words, the $i^{\text{th}}$ column vector $\mathbf{h}_i^p$ (or $\mathbf{h}_i^q$) in $\mathbf{H}^p$ (or $\mathbf{H}^q$) represents the $i^{\text{th}}$ token in the passage (or the question) together with some contextual information from the left.

**Match-LSTM Layer**

We apply the match-LSTM model (Wang & Jiang, 2016) proposed for textual entailment to our machine comprehension problem by treating the question as a premise and the passage as a hypothesis. The match-LSTM sequentially goes through the passage. At position $i$ of the passage, it first uses the standard word-by-word attention mechanism to obtain attention weight vector $\overrightarrow{\alpha}_i \in \mathbb{R}^{1 \times Q}$ as follows:

$$
\begin{aligned}
\overrightarrow{\mathbf{G}}_i &= \tanh(\mathbf{W}^q \mathbf{H}^q + (\mathbf{W}^p \mathbf{h}_i^p + \mathbf{W}^r \overrightarrow{\mathbf{h}}_{i-1}^r + \mathbf{b}^p) \otimes \mathbf{e}_Q), \\
\overrightarrow{\alpha}_i &= \text{softmax}(\mathbf{w}^\mathsf{T} \overrightarrow{\mathbf{G}}_i + b \otimes \mathbf{e}_Q),
\end{aligned}
\tag{2}
$$

where $\mathbf{W}^q, \mathbf{W}^p, \mathbf{W}^r \in \mathbb{R}^{l \times l}, \mathbf{b}^p, \mathbf{w} \in \mathbb{R}^{l \times 1}$ and $b \in \mathbb{R}$ are parameters to be learned, $\overrightarrow{\mathbf{G}}_i \in \mathbb{R}^{l \times Q}$ is the intermediate result, $\overrightarrow{\mathbf{h}}_{i-1}^r \in \mathbb{R}^{l \times 1}$ is the hidden vector of the one-directional match-LSTM (to be explained below) at position $i - 1$, and the outer product $(\cdot \otimes \mathbf{e}_Q)$ produces a matrix or row vector by repeating the vector or scalar on the left for $Q$ times.

Essentially, the resulting attention weight $\overrightarrow{\alpha}_{i,j}$ above indicates the degree of matching between the $i^{\text{th}}$ token in the passage with the $j^{\text{th}}$ token in the question. Next, we use the attention weight vector

---

[4] For the MSMARCO dataset, $\mathbf{P}$ is actually consisted of several unrelated documents. The previous state of pre-processing LSTM and match-LSTM to compute the first state of each document is set to zero.

$\overrightarrow{\alpha}_i$ to obtain a weighted version of the question and combine it with the current token of the passage to form a vector $\overrightarrow{\mathbf{z}}_i$:

$$\overrightarrow{\mathbf{z}}_i = \begin{bmatrix} \mathbf{h}_i^p \\ \mathbf{H}^q \overrightarrow{\alpha}_i^\intercal \end{bmatrix}, \tag{3}$$

where $\mathbf{H}^q \in \mathbb{R}^{l \times Q}$, $\overrightarrow{\alpha}_i \in \mathbb{R}^{1 \times Q}$ and $\mathbf{h}_i^p \in \mathbb{R}^{l \times 1}$. This vector $\overrightarrow{\mathbf{z}}_i$ is fed into a standard one-directional LSTM to form our so-called match-LSTM:

$$\overrightarrow{\mathbf{h}}_i^r = \overrightarrow{LSTM}(\overrightarrow{\mathbf{z}}_i, \overrightarrow{\mathbf{h}}_{i-1}^r), \tag{4}$$

where $\overrightarrow{\mathbf{h}}_i^r \in \mathbb{R}^{l \times 1}$.

We further build a similar match-LSTM in the reverse direction. The purpose is to obtain a representation that encodes the contexts from both directions for each token in the passage.

Let $\overrightarrow{\mathbf{H}}^r \in \mathbb{R}^{l \times P}$ represent the hidden states $[\overrightarrow{\mathbf{h}}_1^r, \overrightarrow{\mathbf{h}}_2^r, \ldots, \overrightarrow{\mathbf{h}}_P^r]$ and $\overleftarrow{\mathbf{H}}^r \in \mathbb{R}^{l \times P}$ represent $[\overleftarrow{\mathbf{h}}_1^r, \overleftarrow{\mathbf{h}}_2^r, \ldots, \overleftarrow{\mathbf{h}}_P^r]$, the hidden states of match-LSTM in the reverse direction. We define $\mathbf{H}^r \in \mathbb{R}^{2l \times P}$ as the concatenation of the two:

$$\mathbf{H}^r = \begin{bmatrix} \overrightarrow{\mathbf{H}}^r \\ \overleftarrow{\mathbf{H}}^r \end{bmatrix}. \tag{5}$$

**Answer Pointer Layer**

The top layer, the Answer Pointer (Ans-Ptr) layer, is motivated by the Pointer Net introduced by Vinyals et al. (2015). This layer uses the sequence $\mathbf{H}^r$ as input. Recall that we have two different models: The *sequence* model produces a sequence of answer tokens but these tokens may not be consecutive in the original passage. The *boundary* model produces only the start token and the end token of the answer, and then all the tokens between these two in the original passage are considered to be the answer. We now explain the two models separately.

**The Sequence Model:** Recall that in the sequence model, the answer is represented by a sequence of integers $\mathbf{a} = (a_1, a_2, \ldots)$ indicating the positions of the selected tokens in the original passage. The Ans-Ptr layer models the generation of these integers in a sequential manner. Because the length of an answer is not fixed, in order to stop generating answer tokens at certain point, we allow each $a_k$ to take up an integer value between 1 and $P + 1$, where $P + 1$ is a special value indicating the end of the answer. Once $a_k$ is set to be $P + 1$, the generation of the answer stops.

In order to generate the $k^{\text{th}}$ answer token indicated by $a_k$, first, the attention mechanism is used again to obtain an attention weight vector $\beta_k \in \mathbb{R}^{1 \times (P+1)}$, where $\beta_{k,j}$ ($1 \le j \le P + 1$) is the probability of selecting the $j^{\text{th}}$ token from the passage as the $k^{\text{th}}$ token in the answer, and $\beta_{k,(P+1)}$ is the probability of stopping the answer generation at position $k$. $\beta_k$ is modeled as follows:

$$\mathbf{F}_k = \tanh(\mathbf{V}\widetilde{\mathbf{H}}^r + (\mathbf{W}^a \mathbf{h}_{k-1}^a + \mathbf{b}^a) \otimes \mathbf{e}_{(P+1)}), \tag{6}$$

$$\beta_k = \text{softmax}(\mathbf{v}^\intercal \mathbf{F}_k + c \otimes \mathbf{e}_{(P+1)}), \tag{7}$$

where $\widetilde{\mathbf{H}}^r \in \mathbb{R}^{2l \times (P+1)}$ is the concatenation of $\mathbf{H}^r$ with a zero vector, defined as $\widetilde{\mathbf{H}}^r = [\mathbf{H}^r; \mathbf{0}]$, $\mathbf{V} \in \mathbb{R}^{l \times 2l}$, $\mathbf{W}^a \in \mathbb{R}^{l \times l}$, $\mathbf{b}^a, \mathbf{v} \in \mathbb{R}^{l \times 1}$ and $c \in \mathbb{R}$ are parameters to be learned, $\mathbf{F}_k \in \mathbb{R}^{l \times (P+1)}$ is the intermediate result, $(\cdot \otimes \mathbf{e}_{(P+1)})$ follows the same definition as before, and $\mathbf{h}_{k-1}^a \in \mathbb{R}^{l \times 1}$ is the hidden vector at position $k - 1$ of an answer LSTM as defined below:

$$\mathbf{h}_k^a = \overrightarrow{LSTM}(\widetilde{\mathbf{H}}^r \beta_k^\intercal, \mathbf{h}_{k-1}^a). \tag{8}$$

We can then model the probability of generating the answer sequence as

$$p(\mathbf{a}|\mathbf{H}^r) = \prod_k p(a_k|a_1, a_2, \ldots, a_{k-1}, \mathbf{H}^r), \tag{9}$$

and

$$p(a_k = j|a_1, a_2, \ldots, a_{k-1}, \mathbf{H}^r) = \beta_{k,j}. \tag{10}$$

To train the model, we minimize the following loss function based on the training examples:

$$-\sum_{n=1}^{N} \log p(\mathbf{a}_n | \mathbf{P}_n, \mathbf{Q}_n). \tag{11}$$

**The Boundary Model:** The boundary model works in a way very similar to the sequence model above, except that instead of predicting a sequence of indices $a_1, a_2, \ldots$, we only need to predict two indices $a_{\mathrm{s}}$ and $a_{\mathrm{e}}$. So the main difference from the sequence model above is that in the boundary model we do not need to add the zero padding to $\mathbf{H}^{\mathrm{r}}$, and the probability of generating an answer is simply modeled as

$$p(\mathbf{a} | \mathbf{H}^{\mathrm{r}}) \quad = \quad p(a_{\mathrm{s}} | \mathbf{H}^{\mathrm{r}}) p(a_{\mathrm{e}} | a_{\mathrm{s}}, \mathbf{H}^{\mathrm{r}}). \tag{12}$$

As this boundary model could point to a span covering too many tokens without any restriction, we try to manually limit the length of the predicted span and then search the span with the highest probability computed by $p(\mathbf{a}_s) \times p(\mathbf{a}_e | \mathbf{a}_s)$ as the answer.

## 3 EXPERIMENTS

In this section, we present our experiment results and perform some analyses to better understand how our models works.

### 3.1 DATA

We use the Stanford Question Answering Dataset (SQuAD) v1.1 and the human-generated Microsoft MAchine Reading COmprehension (MSMARCO) dataset v1.1 to conduct our experiments.

Passages in SQuAD come from 536 articles in Wikipedia covering a wide range of topics. Each passage is a single paragraph from a Wikipedia article, and each passage has around 5 questions associated with it. In total, there are 23,215 passages and 107,785 questions. The data has been split into a training set (with 87,599 question-answer pairs), a development set (with 10,570 question-answer pairs) and a hidden test set.

For the MSMARCO dataset, the questions are user queries issued to the Bing search engine, the context passages are real Web documents and the answers are human-generated. We select the span that has the highest F1 score with the gold standard answer for training and only predict the span in the passages during evaluation. The data has been split into a training set (82326 pairs), a development set (10047 pairs) and a test set (9650 pairs).

### 3.2 EXPERIMENT SETTINGS

We first tokenize all the passages, questions and answers. We use word embeddings from GloVe (Pennington et al., 2014) to initialize the model. Words not found in GloVe are initialized as zero vectors. The word embeddings are not updated during the training of the model.

The dimensionality $l$ of the hidden layers is set to be 150. We use ADAMAX (Kingma & Ba, 2015) with the coefficients $\beta_1 = 0.9$ and $\beta_2 = 0.999$ to optimize the model. Each update is computed through a minibatch of 30 instances. We do not use L2-regularization.

For the SQuAD dataset, the performance is measured by two metrics: percentage of exact match with the ground truth answers and word-level F1 score when comparing the tokens in the predicted answers with the tokens in the ground truth answers. Note that in the development set and the test set each question has around three ground truth answers. F1 scores with the best matching answers are used to compute the average F1 score. For the MSMARCO dataset, the metrics in the official tool of MSMARCO evaluation are BLEU-1/2/3/4 and Rouge-L, which are widely used in many domains.

### 3.3 RESULTS

The SQuAD and MSMARCO results of our models as well as the results of the baselines (Rajpurkar et al., 2016; Yu et al., 2016) are shown in Table 2. For the "LSTM with Ans-Ptr" models, they are

| | SQuAD | | | | MSMARCO |
| | Exact Match | | F1 | | BLEU1/2/3/4 / Rouge-L |
| | Dev | Test | Dev | Test | Dev & Test |
|---|---|---|---|---|---|
| Human | 80.3 | 77.0 | 90.5 | 86.8 | - & 46/-/-/- / 47 |
| Golden Passage | - | - | - | - | 19.6 / 18.8 / 18.1 / 17.5 / 32.3 & - |
| LR (Rajpurkar et al., 2016) | 40.0 | 40.4 | 51.0 | 51.0 | - |
| DCR (Yu et al., 2016) | 62.5 | 62.5 | 71.2 | 71.0 | - |
| LSTM with Ans-Ptr (Sequence) | 37.7 | - | 48.5 | - | 10.3 / 7.2  / 5.6  / 4.6  / 21.6 & - |
| LSTM with Ans-Ptr (Boundary) | 45.2 | - | 55.3 | - | 32.0 / 25.3 / 22.2 / 20.4 / 32.3 & - |
| mLSTM with Ans-Ptr (Sequence) | 54.4 | - | 68.2 | - | 12.5 / 9.2  / 7.5  / 6.5  / 22.5 & - |
| mLSTM with Ans-Ptr (Boundary) | 63.0 | - | 72.7 | - | 32.9 / 26.4 / 23.2 / 21.6 / 33.8 & - |
| Our best boundary model | **67.0** | **66.9** | **77.2** | **77.1** | **40.1 / 33.3 / 30.1 / 28.2 / 37.2** & |
| | | | | | **40.7 / 33.9 / 30.6 / 28.7 / 37.3** |
| mLSTM with Ans-Ptr (Boundary+en) | 67.6 | 67.9 | 76.8 | 77.0 | - |
| Our best boundary model (en) | **71.3** | **72.6** | **80.0** | **80.8** | - |

Table 2: Experiment Results on SQuAD and MSMARCO datasets. Here "LSTM with Ans-Ptr" removes the attention mechanism in match-LSTM (mLSTM) by using the final state of the LSTM for the question to replace the weighted sum of all the states. Our best boundary model is the further tuned model and its ablation study is shown in Table 4. "en" refers to ensemble method.

| | SQuAD #w in A/Q/P | MSMARCO #w in A/Q/P |
|---|---|---|
| raw | 3.1 / 11 / 141 | 16.3 / 6 / 667 |
| seq | 2.4 / - / - | 6.7  / - / - |
| bou | 3.0 / - / - | 15.7 / - / - |

Table 3: Statistical analysis on the development datasets. #w: number of words on average; P: passage; Q: question; A: answer; raw: raw data from the development dataset; seq/bou: the answers generated by the sequence/boundary models with match-LSTM.

| | SQuAD EM & F1 | MSMARCO BLEU1/2/3/4 & Rouge-L |
|---|---|---|
| Best model | 67.0 & 77.2 | 40.1/33.3/30.1/28.2 & 37.2 |
| -bi-Ans-Ptr | 66.5 & 76.8 | 39.9/32.8/29.6/27.9 & 36.7 |
| -deep | 65.9 & 75.8 | 39.6/32.6/29.4/27.4 & 35.9 |
| -elem | 65.2 & 75.4 | 38.1/31.4/28.3/26.5 & 35.5 |
| -pre-LSTM | 64.0 & 72.9 | 39.6/32.8/29.8/27.7 & 36.3 |

Table 4: Ablation study for our best boundary model on the development datasets. Our best model is a further tuned boundary model by considering "bi-Ans-Ptr" which adds bi-directional answer pointer, "deep" which adds another two-layer bi-directional LSTMs between the match-LSTM and the Answer Pointer layers, and "elem" which adds element-wise comparison ,$(\mathbf{h}_i^p - \mathbf{H}^q \alpha_i^\top)$ and $(\mathbf{h}_i^p \odot \mathbf{H}^q \alpha_i^\top)$, into Eqn 3. "-pre-LSTM" refers to removing the pre-processing layer.

the experiments with the ablation of attention mechanism in match-LSTM. Specifically, we use the final representation of the question to replace the weighted sum of the question representations. For the MSMARCO dataset, as the context for each question is consisted of around 10 documents, the "Golden Passage" is to directly use the human labeled document which could answer the question as the prediction.

From the results in Table 2, we can see that the boundary model could clearly outperform the sequence model in a big margin on both datasets. We hypothesis that the sequence model is more likely to stop word generation earlier, and the boundary model can somehow overcome this problem. We have a statistical analysis on the answers generated by our sequence and boundary models shown in Table 3. We can see that the length of the answers generated by the sequence model is much shorter than the ground truth. Especially for the MSMARCO task where the answers are usually much longer, the sequence model could only generate 7 words on average, while the ground truth answers are 16 on average and the boundary model could generate nearly the same number of words with the ground truth. Several answers generated by our models are shown in Appendix A. From Table 2, we can also see that the performance gets poorer by removing the attention mechanism in match-LSTM, while for the MSMARCO dataset, the attention mechanism effects less, with no more than 2 percent reduction in BLEU and Rouge-L scores by attention mechanism ablation.

Based on the effectiveness of boundary pointer and match-LSTM, we conduct further exploration of the boundary model by adding element-wise comparison $(\mathbf{h}_i^p - \mathbf{H}^q \alpha_i^\intercal)$ and $(\mathbf{h}_i^p \odot \mathbf{H}^q \alpha_i^\intercal)$ into Eqn 3 in match-LSTM layer, adding 2 more bi-directional LSTM layers between match-LSTM and Ans-Ptr layers, and adding bi-directional Ans-Ptr. We show the ablation study of this further tuned model in Table 4. We can see that adding element-wise matching could make the biggest improvement for our boundary model. We also try to remove the phrase-level representation by removing the pre-process LSTM and using the word-level representations as the inputs of match-LSTM. Interestingly, we find the phrase-level representation effects little on the MSMARCO task.

Overall, we can see that both of our match-LSTM models have clearly outperformed the logistic regression model by Rajpurkar et al. (2016), which relies on carefully designed features. The improvement of our models over the logistic regression model shows that our end-to-end neural network models without much feature engineering are very effective on these tasks and datasets. Our boundary model also outperformed the DCR model (Yu et al., 2016), which maximizes the probability of the gold standard span from all the candidate spans through a neural network structure.

## 3.4 FURTHER ANALYSES

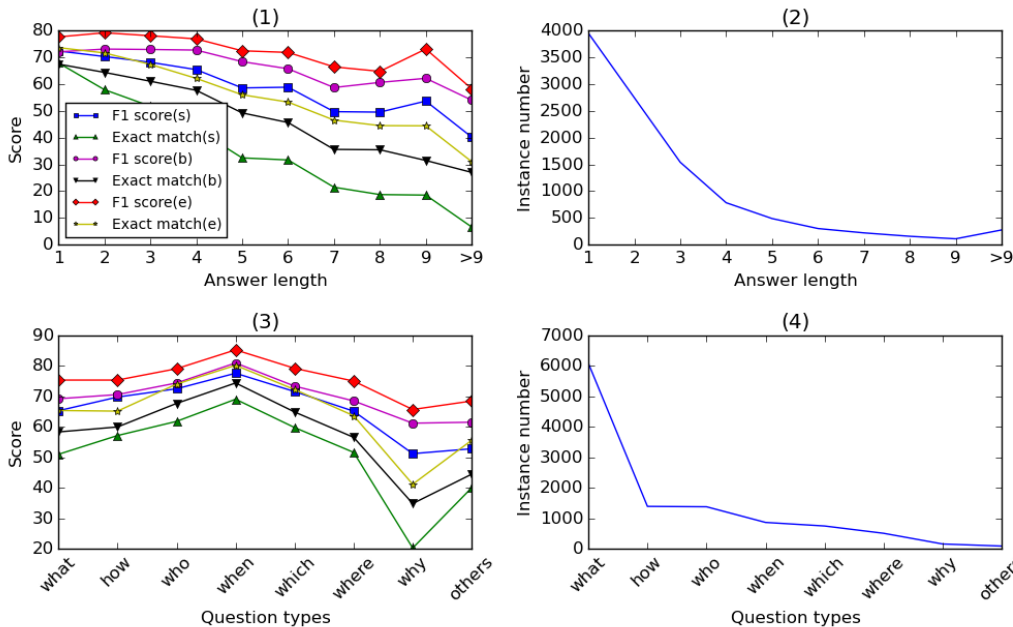

Figure 2: Performance breakdown by answer lengths and question types on SQuAD development dataset. Top: Plot (1) shows the performance of our two models (where *s* refers to the sequence model , *b* refers to the boundary model, and *e* refers to the ensemble boundary model) over answers with different lengths. Plot (2) shows the numbers of answers with different lengths. Bottom: Plot (3) shows the performance our the two models on different types of questions. Plot (4) shows the numbers of different types of questions.

To better understand the strengths and weaknesses of our models, we perform some further analyses of the results below.

First, we suspect that longer answers are harder to predict. To verify this hypothesis, we analysed the performance in terms of both exact match and F1 score with respect to the answer length on the development set, as shown in Figure 2. For example, for questions whose answers contain more than 9 tokens, the F1 score of the boundary model drops to around 55% and the exact match score drops to only around 30%, compared to the F1 score and exact match score of close to 72% and 67%, respectively, for questions with single-token answers. And that supports our hypothesis.

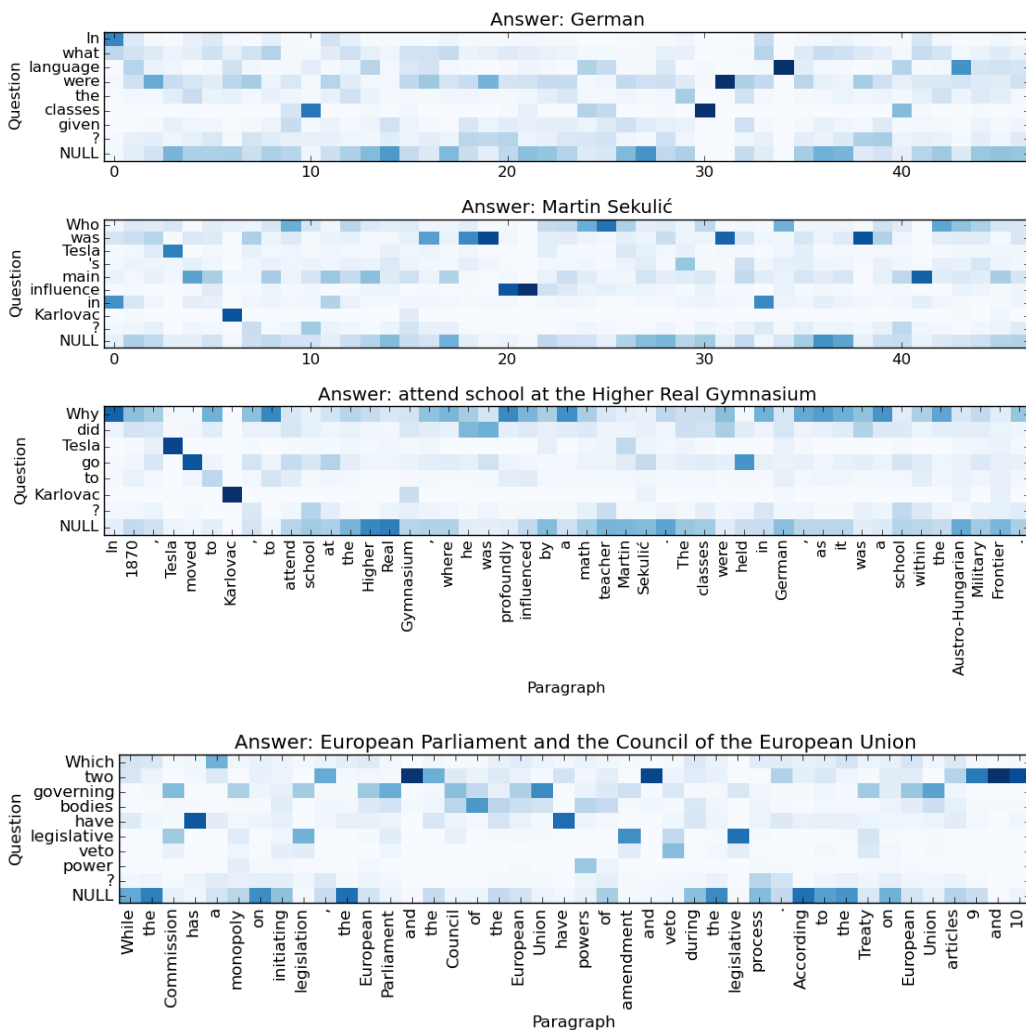

Figure 3: Visualization of the attention weights $\alpha$ for four questions. The first three questions share the same paragraph. The title is the answer predicted by our model.

Next, we analyze the performance of our models on different groups of questions, as shown in Figure 2. We use a crude way to split the questions into different groups based on a set of question words we have defined, including "what," "how," "who," "when," "which," "where," and "why." These different question words roughly refer to questions with different types of answers. For example, "when" questions look for temporal expressions as answers, whereas "where" questions look for locations as answers. According to the performance on the development dataset, our models work the best for "when" questions. This may be because in this dataset temporal expressions are relatively easier to recognize. Other groups of questions whose answers are noun phrases, such as "what" questions, "which" questions and "where" questions, also get relatively better results. On the other hand, "why" questions are the hardest to answer. This is not surprising because the answers to "why" questions can be very diverse, and they are not restricted to any certain type of phrases.

Finally, we would like to check whether the attention mechanism used in the match-LSTM layer is effective in helping the model locate the answer. We show the attention weights $\alpha$ in Figure 3. In the figure the darker the color is the higher the weight is. We can see that some words have been well aligned based on the attention weights. For example, the word "German" in the passage is aligned well to the word "language" in the first question, and the model successfully predicts "German" as the answer to the question. For the question word "who" in the second question, the

word "teacher" actually receives relatively higher attention weight, and the model has predicted the phrase "Martin Sekulic" after that as the answer, which is correct. For the third question that starts with "why", the attention weights are more evenly distributed and it is not clear which words have been aligned to "why". For the last question, we can see that the word knowledge needed for generating the answer can also be detected by match-LSTM. For example, the words "European", "Parliament", "Council", "European" and "Union" have higher attention weights on "governing" in the question. Even though our models can solve this type of questions, they are still not able to solve the questions that need multi-sentences reasoning. More answers generated by our models for the questions related to different kinds of reasoning are shown in Appendix B.

## 4 RELATED WORK

Machine comprehension of text has gained much attention in recent years, and increasingly researchers are building data-drive, end-to-end neural network models for the task. We will first review the recently released datasets and then some end-to-end models on this task.

### 4.1 DATASETS

A number of datasets for studying machine comprehension were created in Cloze style by removing a single token from a sentence in the original corpus, and the task is to predict the missing word. For example, Hermann et al. (2015) created questions in Cloze style from CNN and Daily Mail highlights. Hill et al. (2016) created the Children's Book Test dataset, which is based on children's stories. Cui et al. (2016) released two similar datasets in Chinese, the People Daily dataset and the Children's Fairy Tale dataset.

Instead of creating questions in Cloze style, a number of other datasets rely on human annotators to create real questions. Richardson et al. (2013) created the well-known MCTest dataset and Tapaswi et al. (2016) created the MovieQA dataset. In these datasets, candidate answers are provided for each question. Similar to these two datasets, the SQuAD dataset (Rajpurkar et al., 2016) was also created by human annotators. Different from the previous two, however, the SQuAD dataset does not provide candidate answers, and thus all possible subsequences from the given passage have to be considered as candidate answers.

Besides the datasets above, there are also a few other datasets created for machine comprehension, such as WikiReading dataset (Hewlett et al., 2016) and bAbI dataset (Weston et al., 2016), but they are quite different from the datasets above in nature.

### 4.2 END-TO-END NEURAL NETWORK MODELS FOR MACHINE COMPREHENSION

There have been a number of studies proposing end-to-end neural network models for machine comprehension. A common approach is to use recurrent neural networks (RNNs) to process the given text and the question in order to predict or generate the answers (Hermann et al., 2015). Attention mechanism is also widely used on top of RNNs in order to match the question with the given passage (Hermann et al., 2015; Chen et al., 2016). Given that answers often come from the given passage, Pointer Network has been adopted in a few studies in order to copy tokens from the given passage as answers (Kadlec et al., 2016; Trischler et al., 2016). Compared with existing work, we use match-LSTM to match a question and a given passage, and we use Pointer Network in a different way such that we can generate answers that contain multiple tokens from the given passage.

Memory Networks (Weston et al., 2015) have also been applied to machine comprehension (Sukhbaatar et al., 2015; Kumar et al., 2016; Hill et al., 2016), but its scalability when applied to a large dataset is still an issue. In this work, we did not consider memory networks for the SQuAD/MSMARCO datasets.

The setting of visual question answering (Antol et al., 2015) is quite similar to machine comprehension, while their answers are usually very short. So the sequence order of the word-level attention representation used to align the figure and the question(Xu & Saenko, 2016; Fukui et al., 2016; Lu et al., 2016), are not used in VQA. While our model focus on the word-by-word attention and use

LSTM to concatenate the aligned pairs and that would be helpful to generate a longer sequence as answer.

## 5 CONCLUSIONS

In this paper, We developed two models for the machine comprehension problem defined in the Stanford Question Answering (SQuAD) and A Human-Generated MAchine Reading COmprehension (MSMARCO) datasets, both making use of match-LSTM and Pointer Network. Experiments on the SQuAD and MSMARCO datasets showed that our second model, the boundary model, could achieve a performance close to the state-of-the-art performance on the SQuAD dataset and achieved the state-of-the-art on the MSMARCO dataset. We also show the boundary model could overcome the early stop prediction problem of the sequence model.

In the future, we plan to look further into the different types of questions and focus on those questions which currently have low performance, such as the "why' questions and multi-sentences related questions. We also plan to test how our models could be applied to other machine comprehension datasets.

## 6 ACKNOWLEDGMENTS

This research is supported by the National Research Foundation, Prime Ministers Office, Singapore under its International Research Centres in Singapore Funding Initiative.

We thank Pranav Rajpurkar for testing our model on the hidden test dataset and Percy Liang for helping us with the Dockerfile for Codalab.

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

# A   APPENDIX

We show the predictions our boundary and sequence models on two cases from two datasets in Table 5. It can be seen that the sequence model is more likely to predict a shorter sequence which is the problem of early stop prediction.

| | |
|---|---|
| **(1) Context** | Asopposed to broadcasts of primetime series, CBS broadcast special episodes of its late night talk shows as its lead-out programs for Super Bowl 50, beginning with a special episode of **The Late Show with Stephen Colbert** following the game. |
| **Question (Syntactic)** | What CBS show followed the Super Bowl? |
| **Golden Anwser** | The Late Show with Stephen Colbert |
| match-LSTM (Sequence) | The Late Show |
| match-LSTM (Boundary) | The Late Show with Stephen Colbert |
| **(2) Context** | Urinalysis is a test that evaluates a sample of your urine. Urinalysis is used to **detect and assess a wide range of disorders, such as urinary tract infection, kidney disease and diabetes.** Urinalysis involves examining the appearance, concentration and content of urine. Abnormal urinalysis results may point to a disease or illness. For example, a urinary tract infection can make urine look cloudy instead of clear. Increased levels of protein in urine can be a sign of kidney disease. |
| **Query** | what can urinalysis detect? |
| **Golden Anwser** | Detect and assess a wide range of disorders, such as urinary tract infection, kidney disease and diabetes. |
| match-LSTM (Sequence) | Urinalysis |
| match-LSTM (Boundary) | Urinalysis is used to detect and assess a wide range of disorders, such as urinary tract infection, kidney disease and diabetes |

Table 5: Prediction samples for sequence and boundary models. The first case is sampled from SQuAD dataset and the second is sampled from MSMARCO dataset.

# B  APPENDIX

We show how four different models work on different type of questions in SQuAD dataset through Table 6. After the analysis of a hundred cases, we see that our models are not able to solve the questions that need multi-sentences reasoning. And the model without attention mechanism has less power to identify the important key word like the third case shown in Table 6.

| | |
|---|---|
| **(1) Context** | The Rankine cycle is sometimes referred to as a **practical Carnot cycle** because, when an efficient turbine is used, the TS diagram begins to resemble the Carnot cycle. |
| **Question (Synonymy)** | What is the Rankine cycle sometimes called? |
| **Golden Anwser** | practical Carnot cycle |
| LSTM (Sequence) | Carnot cycle |
| match-LSTM (Sequence) | Carnot cycle |
| LSTM (Boundary) | practical Carnot cycle |
| match-LSTM (Boundary) | Carnot cycle |
| **(2) Context** | While the Commission has a monopoly on initiating legislation, **the European Parliament and the Council of the European Union** have powers of amendment and veto during the legislative process. |
| **Question (Knowledge)** | Which two governing bodies have legislative veto power? |
| **Golden Anwser** | the European Parliament and the Council of the European Union |
| LSTM (Sequence) | European Parliament and the Council of the European Union |
| match-LSTM (Sequence) | European Parliament and the Council of the European Union |
| LSTM (Boundary) | European Parliament and the Council of the European Union |
| match-LSTM (Boundary) | European Parliament and the Council of the European Union |
| **(3) Context** | Current faculty include the anthropologist Marshall Sahlins, historian Dipesh Chakrabarty, ... Shakespeare scholar **David Bevington**, and renowned political scientists John Mearsheimer and Robert Pape. |
| **Question (Syntactic)** | What Shakespeare scholar is currently on the university's faculty? |
| **Golden Anwser** | David Bevington |
| LSTM (Sequence) | Marshall Sahlins |
| match-LSTM (Sequence) | David Bevington |
| LSTM (Boundary) | Marshall Sahlins |
| match-LSTM (Boundary) | David Bevington |
| **(4) Context** | The V&A Theatre & Performance galleries, formerly the Theatre Museum, opened in March 2009. The collections are stored by the V&A, and are available for research, exhibitions and other shows. They hold the UK's biggest national collection of **material about live performance** in the UK since Shakespeare's day, covering drama, dance, musical theatre, circus, music hall, rock and pop, and most other forms of live entertainment. |
| **Question (Reasoning)** | What collection does the V&A Theatre & Performance galleries hold? |
| **Golden Anwser** | material about live performance |
| LSTM (Sequence) | Theatre |
| match-LSTM (Sequence) | the Theatre Museum |
| LSTM (Boundary) | research, exhibitions and other shows |
| match-LSTM (Boundary) | Theatre Museum |
| **(5) Context** | Along with giving the offender his "just deserts", **achieving crime control via incapacitation** and deterrence is a major goal of criminal punishment. |
| **Question (Ambiguous)** | What is the main goal of criminal punishment of civil disobedients? |
| **Golden Anwser** | achieving crime control via incapacitation and deterrence |
| LSTM (Sequence) | deterrence |
| match-LSTM (Sequence) | just deserts |
| LSTM (Boundary) | incapacitation and deterrence |
| match-LSTM (Boundary) | incapacitation and deterrence |

Table 6: Different types of reasoning samples in SQuAD dataset. "match-LSTM" refers to the "match-LSTM with Ans-Ptr" and "LSTM" refers to the "LSTM with Ans-Ptr" which is the ablation of attention mechanism in match-LSTM.

