# Peer review of "Machine Comprehension Using Match-LSTM and Answer Pointer"

_ICLR 2017 — accepted_

[Author Response · Shuohang Wang · 13 Dec 2016]
**An updated pdf version**

Dear reviewers,

Thank you for your valuable comments again! We have made the corresponding revisions and updated a new pdf version. We briefly list the changes here:

AnonReviewer1:
1.We clarify the dimension of row vector $\alpha_i$ to be $1\times Q$.
2.We add the visualization of the $\alpha$ values for the question requiring world knowledge in Figure 2 and add the corresponding analysis at the end of the section "Experiments".

AnonReviewer2:
We revise the description of the state-of-the-art results in the last paragraph in "Introduction".
We clarify the dimension of $G$ to be $l\time Q$, the row vector $\alpha_i$ to be $1\times Q$, the column vector $w$ to be $l\times 1$ for equation (2). So is the equation (8).
We clarify the statement of footnote 3 about the output gates in the pre-processing layer.
We clarify the description of global search on the spans in both the boundary model description part and the Table 2.

AnonReviewer3:
We clarify the integration of match-LSTM and pointer network in the last two paragraphs of the "Introduction".
We directly cite the works of the baselines in Table 2.

Thanks,
Shuohang

[Official Review · AnonReviewer2 · rating 6 · confidence 3 · 16 Dec 2016]
**No Title**

SUMMARY.
This paper proposes a new neural network architectures for solving the task of reading comprehension question answering where the goal is answering a questions regarding a given text passage.
The proposed model combines two well-know neural network architectures match-lstm and pointer nets.
First the passage and the questions are encoded with a unidirectional LSTM.
Then the encoded words in the passage and the encoded words in the questions are combined with an attention mechanism so that each word of the passage has a certain degree of compatibility with the question.
For each word in the passage the word representation and the weighted representation of the query is concatenated and passed to an forward lstm.
The same process is done in the opposite direction with a backward lstm.
The final representation is a concatenation of the two lstms.
As a decoded a pointer network is used.
The authors tried with two approaches: generating the answer word by word, and generating the first index and the last index of the answer.

The proposed model is tested on the Stanford Question Answering Dataset.
An ensemble of the proposed model achieves performance close to state-of-the-art models.


----------

OVERALL JUDGMENT

I think the model is interesting mainly because of the use of pointer networks as a decoder.
One thing that the authors could have tried is a multi-hop approach. It has been shown in many works to be extremely beneficial in the joint encoding of passage and query. The authors can think of it as a deep match-lstm.
The analysis of the model is interesting and insightful.
The sharing of the code is good.

[Official Review · AnonReviewer3 · rating 6 · confidence 4 · 17 Dec 2016]
**Review: Interesting combination of existing approaches with encouraging results**

The paper looks at the problem of locating the answer to a question in a text (For this task the answer is always part of the input text). For this the paper proposes to combine two existing works: Match-LSTM to relate question and text representations and Pointer Net to predict the location of the answer in the text.

Strength:
-	The suggested approach makes sense for the task and achieves good performance, (although as the authors mention, recent concurrent works achieve better results)
-	The paper is evaluated on the SQuAD dataset and achieves significant improvements over prior work.


Weaknesses:
1.	It is unclear from the paper how well it is applicable to other problem scenarios where the answer is not a subset of the input text.
2.	Experimental evaluation
2.1.	It is not clear why the Bi-Ans-Ptr in Table 2 is not used for the ensemble although it achieves the best performance.
2.2.	It would be interested if this approach generalizes to other datasets.


Other (minor/discussion points)
-	The task and approach seem to have some similarity of locating queries in images and visual question answering. The authors might want to consider pointing to related works in this direction.
-	I am wondering how much this task can be seen as a “guided extractive summarization”, i.e. where the question guides the summarization process.
-	Page 6, last paragraph: missing “.”: “… searching This…”



Summary:
While the paper presents an interesting combination of two approaches for the task of answer extraction, the novelty is moderate. While the experimental results are encouraging, it remains unclear how well this approach generalizes to other scenarios as it seems a rather artificial task.

[Official Review · AnonReviewer1 · rating 7 · confidence 3 · 17 Dec 2016 (modified: 20 Jan 2017)]
**More analyses / ablation studies / insights needed regarding the functioning of the proposed model**

Summary:
The paper presents a deep neural network for the task of machine comprehension on the SQuAD dataset. The proposed model is based on two previous works -- match-LSTM and Pointer Net. Match-LSTM produces attention over each word in the given question for each word in the given passage, and sequentially aggregates this matching of each word in the passage with the words in the question. The pointer net is used to generate the answer by either generating each word in the answer or by predicting the starting and ending tokens in the answer from the provided passage. The experimental results show that both the variants of the proposed model outperform the baseline presented in the SQuAD paper. The paper also shows some analysis of the results obtained such as variation of performance across answer lengths and question types.

Strengths:
1. A novel end-to-end model for the task of machine comprehension rather than using hand-crafted features.
2. Significant performance boost over the baseline presented in the SQuAD paper.
3. Some insightful analyses of the results such as performance is better when answers are short, "why" questions are difficult to answer.

Weaknesses/Questions/Suggestions:
1. The paper does not show quantitatively how much modelling attention in match-LSTM and answer pointer layer helps. So, it would be insightful if authors could compare the model performance with and without attention in match-LSTM, and with and without attention in answer pointer layer.
2. It would be good if the paper could provide some insights into why there is a huge performance gap between boundary model and sequence model in the answer pointer layer.
3. I would like to see the variation in the performance of the proposed model for questions that require different types of reasoning (table 3 in SQuAD paper). This would provide insights into what are the strengths and weaknesses of the proposed model w.r.t the type reasoning required.
4. Could authors please explain why the activations resulting from {h^p}_i and {h^r}_{i-1} in G_i in equation 2 are being repeated across dimension of Q. Why not learn different activations for each dimension? 
5. I wonder why Bi-Ans-Ptr is not used in the ensemble model (last row in table 2) when it is shown that Bi-Ans-Ptr improves performance by 1.2% in F1.
6. Could authors please discuss and compare the DCR model (in table 2) in the paper in more detail?

Review Summary: The paper presents a reasonable end-to-end model for the task of machine comprehension on the SQuAD dataset, which outperforms the baseline model significantly. However, it would be good if more analyses / ablation studies / insights are included regarding -- how much attention helps, why is boundary model better than sequence model, how does the performance change when the reasoning required becomes difficult.

[Final Decision · Program Chairs · 06 Feb 2017]
**ICLR committee final decision**

This paper provides two approaches to question answering: pointing to spans, and use of match-LSTM. The models are evaluated on SQuAD and MSMARCO. The reviewers we satisfied that, with the provision of additional comparisons and ablation studies submitted during discussion, the paper was acceptable to the conference, albeit marginally so.